# Association between occupational testicular radiation exposure and lower male sex ratio of offspring among orthopedic surgeons

Yasukazu Hijikata[1,2]*, Masayuki Nakahara[1], Akira Kusumegi[3], Junji Morii[1‡], Naoki Okubo[1‡], Nozomi Hatano[1‡], Yuichi Takahashi[3‡]

1 Department of Spine and Low Back Pain Center, Kitasuma Hospital, Hyogo, Japan, 2 Department of Healthcare Epidemiology, School of Public Health in the Graduate School of Medicine, Kyoto University, Kyoto, Japan, 3 Department of Spine and Spine Surgery, Shinkomonji Hospital, Fukuoka, Japan

☯ These authors contributed equally to this work.
‡ These authors also contributed equally to this work
* hiz007@hotmail.co.jp

## Abstract

### Background

Exposure to occupational radiation can lower the male sex ratio. However, specific radiation exposure to the testes has not been evaluated.

### Objective

This study aimed to examine the association between testicular radiation exposure and lower male sex ratio in children.

### Methods

A comprehensive questionnaire survey was administered to 62 full-time male doctors with children aged < 10 years at 5 hospitals. Based on the possibility of testicular radiation exposure 1 year before the child's birth, participants were assigned to 3 groups as follows: RT (orthopedic surgery), RNT (cardiology/neurosurgery), and N (others). Intergroup differences in the proportion of female children were ascertained, and the female sex ratio (number of female/total number) of each group was compared against the standard value of 0.486. Multivariate logistic regression analysis with a generalized estimating equation was used to model the effects on the probability of female birth while controlling for the correlation among the same fathers.

### Results

The study population included 62 fathers and 109 children, 49 were female: 19/27, 11/30, and 19/52 in the RT, RNT, and N group, respectively; the RT group had the highest proportion of females ($p = 0.009$). The $p$ values for comparisons with the standard sex ratio (0.486) were 0.02, 0.19, and 0.08 for the RT, RNT, and N groups, respectively. Based on the N

**Data Availability Statement:** All relevant data are within the manuscript and its Supporting Information files.

**Funding:** The author(s) received no specific funding for this work.

**Competing interests:** The authors have declared that no competing interests exist.

group, the adjusted odds ratios for the child to be female were 4.40 (95% confidence interval 1.60–2.48) and 1.03 (0.40–2.61) for the RT and RNT groups, respectively.

## Conclusions

Our results imply an association between testicular radiation exposure and low male sex ratio of offspring. Confirmatory evidence is needed from larger studies which measure the pre-conceptional doses accumulated in various temporal periods, separating out spermato-gonial and spermatid effects.

## Introduction

The number of radiation-generating medical procedures has increased over time, especially for spinal surgery. It includes minimally invasive procedures, such as lateral lumbar interbody fusion and percutaneous pedicle screw insertion. Thus, there have been concerns on the compounded radiation exposure of surgeons [1–4]. Although several clinical studies have reported that doses of exposure are within the recommended limits, the current radiation exposure limits are extrapolated from the values reported in studies of individuals who survived the atomic bombings in Hiroshima and Nagasaki [5–7]. Thus, the limited evidence on the long-term health-related effects of low-dose radiation raises concerns on the development of cataract, thyroid diseases, and cancer [8–12].

Despite an increased interest in protection against radiation, there is insufficient motivation among orthopedic surgeons to assess the use of protection against radiation because the direct effects of radiation exposure are not tangible and may not be discernible for several years [13, 14]. Thus, there is an explicit need to induce evidence-based and awareness-based behavioral changes to urge orthopedic surgeons to take action to avoid radiation exposure.

This study focused on the association between radiation exposure and the sex ratio of the offspring as an indicator of the potential effect of radiation exposure. We hypothesized that testicular radiation exposure skews the sex ratio of the offspring of male doctors toward the female sex by decreasing the ratio of sex-determining sperms. Occupational radiation exposure has been previously suggested to lower the male sex ratio of the offspring of male radiologists, male orthopedic surgeons, and young male cardiologists [15–17]. However, these studies did not evaluate testicular radiation exposure. Moreover, previous studies administered questionnaire surveys where non-responder-induced selection bias is a major limitation. Therefore, we conducted this study to examine the association between testicular radiation exposure and higher female sex ratio using a comprehensive survey of at-risk groups.

## Materials and methods

### Study design and setting

We conducted a case-control study using data obtained from a comprehensive survey of full-time doctors at 5 private hospitals in the Kyushu region, Japan—Shinkomonji Hospital, Fukuokawajiro Hospital, Shinyukuhashi Hospital, Shinmizumaki Hospital, and Shintakeo Hospital. Each study center was a designated emergency care and training hospital. Therefore, these hospitals have the unique advantage of being provided with many opportunities for treating trauma cases and employing many young, full-time doctors. The survey was conducted in February 2017, with the permission of the director of each hospital. The publication of the

results was approved by the certified ethics committee of the main research institution, Shin-komonji Hospital. Information was collected from the study participants via individual inter-views, conducted by medical clerks. The information was anonymized and sent to the researchers. Provision of responses to the survey was regarded as provision of tacit consent for study participation; the opt-out consent process was used.

### Study population and data collection

The study population comprised all full-time male doctors with at least one child aged <10 years at the time of the survey. In consideration of the psychological burden of the respon-dents, details of any deceased children were not included. The exclusion criterion was refusal for study participation. The survey was conducted in February 2017 by the secretary of the medical office in each hospital. Physicians with more than one child were counted more than once; therefore, the number of participants included in the analysis is the number of children, not the number of physicians. There were two rationales for limiting the age of the children to 10 years: one was to reduce recall bias, and the other was to limit the study inclusion eligibility to at-risk groups. Because of the increasing medical radiation exposure over time [1], the younger generation is more likely to be exposed to radiation. Moreover, younger and less experienced physicians tend to have higher radiation exposure [5].

### Outcome of interest and exposure

The outcome of interest was defined as having a female child. The participants were assigned to three groups based on the departments they belonged to a year before the birth of the child, which were as follows: departments that used medical radiation and had a high possibility of testicular radiation exposure (RT group), departments that used medical radiation but had a low possibility of testicular radiation exposure (RNT group), and departments that did not use medical radiation, including medical students (N group). Particularly, the RT group included doctors from the department of orthopedic surgery, wherein radiation is used primarily dur-ing surgery, whereas the RNT group included doctors from cardiology/neurosurgery, wherein radiation is used during catheterization.

In all study centers, apron-type protectors were generally used in the operating room, and coat-type protectors were used in the catheterization room. A coat-type protector provides full circumferential coverage of the torso, whereas the apron-type protector covers only the front of the torso. Thus, using apron-type protectors leave the testes unprotected against lateral and posterior radiation exposure. In addition, the physicians primarily face the source of the radia-tion in the catheterization room, whereas the surgeons deliver radiation at various angles and positions in the operating room. Therefore, we speculated that orthopedic surgeons are at a greater risk of experiencing lateral and posterior radiation exposure that could potentially affect the testes (Fig 1). The radiation dose to the testes considered herein is similar to that in the study by Funao et al. [18]. They reported that during a single interbody fusion, the average radiation exposure of an unshielded surgeon to scattered radiation in the genital area was 0.15 mSv, whereas the average radiation exposure of a radiographer away from the surgical field was 0.05 mSv—a difference of approximately 0.1 mSv/procedure.

The surgeon wore an apron-type protector, leaving the lateral and posterior torso unpro-tected. The radiation source of the anteroposterior fluoroscopy was on the floor to ensure that the lower half of the torso is exposed to scattered rays.

The rationale for the selection of a cut-off date of exactly 1 year before the birth of the child was to localize the period at risk of changes that led to a lower male sex ratio. It takes approxi-mately 74 days for the spermatogonia to differentiate and mature into sperm [19]. We

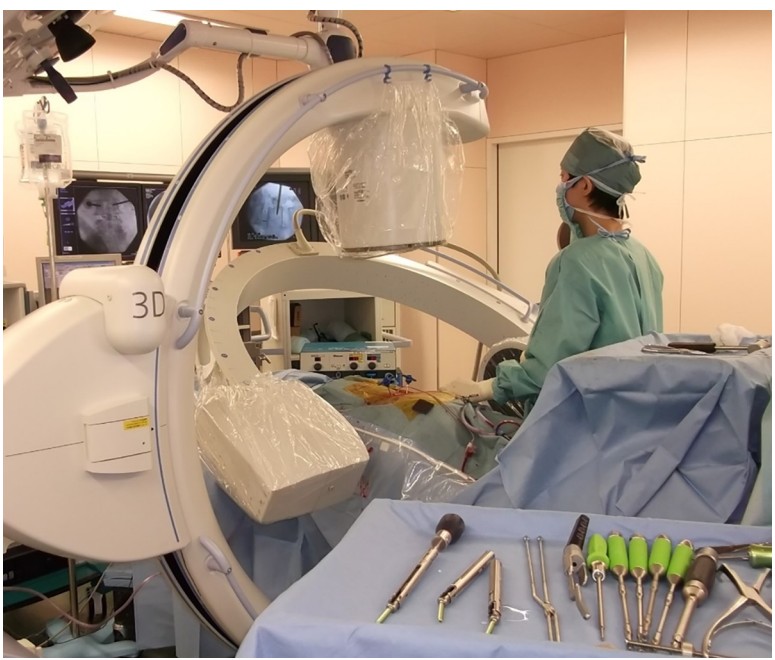

**Fig 1. Percutaneous pedicle screw insertion in a two-way C-arm position.**

hypothesized, albeit without supporting evidence, that testicular radiation exposure reduces the male sex ratio of the child by decreasing the ratio of sex-determining sperms. Accordingly, we estimated that the period at risk was 2–3 months before conception. This duration is related to the period after which the primary spermatocyte has differentiated into the secondary 'male' and 'female' spermatocytes and the period when the mature sperm has migrated to the epididymis where it is stored. In other words, considering that the gestation period is approximately 280 days, the period at risk of a decrease in the male sex ratio of the child due to testicular radiation exposure began approximately one year before the child's birthday.

## Statistical analyses

We described the paternal age at the child's conception and the proportion of female children by the paternal exposure status. We used Fisher's exact test to compare the proportion of female children between groups. A single-sample test of proportion was used to compare the observed female sex ratio (number of female children/total number of children) of each group with the standard ratio of 0.486 in Japan [20]. Furthermore, multivariate logistic regression analysis was conducted to model the effects of the exposure variables (RT and RNT) on the probability of having a female child. We used generalized estimating equations as a control for the correlation of observations among the same fathers. We included the paternal age at the child's conception in the analytical model. Statistical analyses were performed using Stata version 15.1 (Stata Corporation, College Station, TX, USA).

## Results

The study population comprised 62 fathers and 109 children, and the responses were obtained from all eligible doctors (response rate, 100%). There were 27, 30, and 52 participants in the RT, RNT, and N groups, respectively. There was no observable difference in paternal age at the child's conception (Table 1). Overall, 49 children (45%) were female: 19 (70%) in the RT

**Table 1. Paternal age at the child's conception and the proportion of female children.**

| | Total, N = 109 | Paternal exposure status | | | P value |
|---|---|---|---|---|---|
| | | Nᵃ, n = 52 (48) | RTᵇ, n = 27 (25) | RNTᶜ, n = 30 (28) | |
| Paternal age at the child's conception, years | 35.2 (6.1) | 36.2 (7.3) | 33.3 (3.9) | 35.1 (5.3) | |
| Proportion of female children | 49 (45) | 19 (37) | 19 (70) | 11 (37) | 0.009 |

Data are presented as means (standard deviations) and numbers (%).

[a]N, physicians working in departments that did not use medical radiation, including medical students.

[b]RT, physicians working in departments that used medical radiation and had a high possibility of testicular radiation exposure.

[c]RNT, physicians working in departments that used medical radiation but had a low possibility of testicular radiation exposure.

group, 11 (37%) in the RNT group, and 19 (37%) in the N group. A significantly higher proportion of females was observed in the RT group than in the other groups ($p$ = 0.009).

The female sex ratios (number of female children/total number of children) were 0.70, 0.37, and 0.37 for the RT, RNT, and N groups, respectively (Table 2). The two-sided $p$ values for the comparison with the national standard value of 0.486 were 0.02, 0.19, and 0.08 for the RT, RNT, and N groups, respectively.

The crude odds ratios for the child to be female on the basis of the ratio in the N group were 4.00 (95% confidence interval [CI], 1.50–10.7) and 0.99 (95% CI, 0.39–2.48) for the RT and RNT groups, respectively, with adjusted odds ratios of 4.40 (95% CI, 1.60–12.1) and 1.03 (95% CI, 0.40–2.61), respectively (Table 3).

## Discussion

### Key results and clinical implications

The results of a survey of 5 private hospitals suggested that working in the department of orthopedic surgery—which entailed a high chance of testicular radiation exposure—at the time of conception was associated with a lower sex ratio, i.e., increase in female offspring (crude odds ratio 4.0, adjusted odds ratio 4.4). These results support the hypothesis that testicular radiation exposure skews the sex ratio of offspring towards females and motivate an investigation of the association between occupational testicular radiation exposure and lower sex ratio. The younger generation, who might be exposed to radiation more frequently, is also the

**Table 2. Female sex ratio of each paternal status and comparison with the national standard value.**

| Paternal exposure status | Female sex ratio of the childᵃ | P valueᵇ |
|---|---|---|
| Total | 0.45 (0.36–0.54) | 0.45 |
| Nᶜ | 0.37 (0.23–0.50) | 0.08 |
| RTᵈ | 0.70 (0.53–0.88) | 0.02 |
| RNTᵉ | 0.37 (0.19–0.54) | 0.19 |

Data are presented as ratios (95% confidence intervals).

[a]Sex ratio was calculated as the number of female children divided by the total number of children.

[b]$P$ values were calculated using a single-sample test of proportion based on the Japanese standard value of 0.486.

[c]N, physicians working in departments that did not use medical radiation, including medical students.

[d]RT, physicians working in departments that used medical radiation and had a high possibility of testicular radiation exposure.

[e]RNT, physicians working in departments that used medical radiation but had a low possibility of testicular radiation exposure.

**Table 3. Crude and adjusted odds ratios of having a female child of the RT and RNT groups.**

| Paternal exposure status | N | Unadjusted OR (95% CI) | P value | Adjusted OR (95% CI)[a] | P value |
|---|---|---|---|---|---|
| N[b] | 52 | Reference | — | Reference | — |
| RT[c] | 27 | 4.00 (1.50, 10.7) | 0.006 | 4.40 (1.60, 12.1) | 0.004 |
| RNT[d] | 30 | 0.99 (0.39, 2.48) | 0.98 | 1.03 (0.40, 2.61) | 0.96 |

OR, odds ratio; CI, confidence interval.

[a]Adjusted OR values were estimated from a logistic regression model adjusted for paternal age at conception.

[b]N, physicians working in departments that did not use medical radiation, including medical students.

[c]RT, physicians working in departments that used medical radiation and had a high possibility of testicular radiation exposure.

[d]RNT, physicians working in departments that used medical radiation but had a low possibility of testicular radiation exposure.

generation that is likely to have children; thus, sex ratio may be of great concern to them. The results of this study contribute to raising awareness on radiation exposure among the younger generation and promoting protection against radiation.

## Comparison with other studies

Our results are consistent with the inconclusive evidence on occupational exposure of healthcare workers to medical X-rays. Hama et al. [15] administered a questionnaire survey among 700 male radiologists working in university hospitals, with a response rate of 89%, and reported a low percentage of male children (48.5%). Interestingly, upon limiting the subjects to those who responded "yes" to the question "Have you ever received a dose of radiation higher than the maximum recommended by the International Commission on Radiological Protection?", the percentage of male children was 34.5% (30/87), which is similar to what we identified in the RT group (30%). Zadeh et al. [16] conducted a questionnaire survey among 504 orthopedic surgeons (response rate, 66%) and reported that 47.0% of their children were male. Pillarisetti et al. [17] conducted a web-based survey of 8,000 cardiologists (response rate, 6%) and noted a similar trend, with 49.9% of the children being male. However, these results are limited by selection biases because they used questionnaires with voluntary responses.

The effect of radiation exposure other than medical X-ray exposure on the sex ratio of the offspring of healthcare workers remains controversial. Jablon & Kato [21] studied 1172 Japanese men exposed in utero to the atomic bomb and reported that exposure had no effect on the sex ratio of their children. Shea & Little [22] also found that in a study of 7678 children, in which a questionnaire was used to determine whether the father had been X-rayed in the 12 months prior to conception, the father's exposure had no effect on the sex ratio of the child. In a study of nearly 150,000 children in India, Koya et al. [23] observed no difference in the sex ratio of children born in natural high-level radiation areas and those in natural normal-level radiation areas. Moreover, Dickinson et al. [24] reported a sex ratio (number of male children/number of female children) of 1.101 in a study of approximately 10,000 children whose fathers had worked in a nuclear facility before conception. Scherb et al. [25] found a sharp increase in the male birth rate after 1986 in Russia and Cuba; Cuba imported 60% of their food from the Soviet Union and attributed this finding to radioactive contamination caused by the Chernobyl disaster. In particular, the results of the two studies cited above are inconsistent with our findings. The discrepancy in results may be due to differences in various factors (e.g., source dose, frequency of exposure, continuous versus intermittent exposure, external versus internal exposure, and amount of testicular exposure) that may have affected the sex ratio of offspring.

## Possible explanations of our findings

It is unclear whether the Y chromosome is more radiosensitive than the X chromosome, which could result in a higher concentration of X-chromosome-bearing sperm (X-CBS) in the testes and thereby a higher probability of female births that could decrease the sex ratio. However, a previous study [26] revealed that male mice exposed to competitive risks had lower proportion of Y-CBS. A study [27] in dogs indicated that the ingestion of environmental toxicants might cause bias in the sex ratio of the sperm toward X-CBS. Various factors have been suggested to reduce the sex ratio of sperms in non-human mammals.

As mentioned earlier, physicians perform operations while facing the table with a coat-type protector in the catheterization room, whereas surgeons perform surgery in a variety of positions with an apron-type protector in the operating room. The position of the surgeon is strongly associated with radiation exposure, as chest exposure at a 90˚ angle to the source position was reported to have twice as high radiation exposure as that in the forward-facing position [28]. Because lead protectors have a near-100% shielding effect [28, 29], the amount of radiation exposure varies greatly between the areas covered by the protector and those that are exposed. Therefore, it was clear that the RT group possessed a higher risk for testicular radiation exposure than the RNT group. In this study, the sex ratio in the RT group was notably lower, whereas that in the RNT group was similar to that in the N group. These results imply that testicular radiation exposure could possibly reduce the sex ratio of the sperm.

## Strengths and limitations

A key strength of our study was the use of complete enumeration data of at-risk populations. A comprehensive survey allowed us to present associations without non-respondent bias.

However, this study had several limitations. First, possible radiation exposure might have been misclassified, and to date, there is no available measurement for actual testicular radiation exposure. Because radiation exposure is influenced by various factors, such as the radiation dose and surgeon's protective actions, this poses a limitation to defining testicular radiation exposure based solely on information on the department to which the father belonged. Furthermore, department affiliation was only available at one time point, i.e., one year before the birth of the child. The assumption of the period at risk could have led to misclassification if the personnel's department had changed within that period. In addition, because we could not determine the timing of exposure, we cannot differentiate between spermatid effects, which we hypothesized, and spermatogonial effects. Second, there was the possible presence of unmeasured confounders, as we have only adjusted for paternal age at the child's conception. Ideally, a randomized controlled trial would provide a more valid comparison. However, it is ethically impossible to enforce and verify this hypothesis. Furthermore, no strong risk factors affecting the sex ratio of the offspring have been reported, and it can be inferred that the impact of unmeasured confounders is small [30]. Third, because the outcomes in this study are not rare, the odds ratios cannot be approximated to risk ratios (i.e., they cannot indicate the strength of the risk of the RT group). Thus, the odds ratios reported in this study should be interpreted with caution, as they are only a measure of the presence or absence of an association. Finally, although the results were highly statistically significant, they are based on a quite small sample, and the play of chance, or some unidentified confounding factor, cannot be ruled out. The fact that the sex ratio was higher in the RNT and N groups might imply a chance effect. In addition, evidence for a causal relationship between testicular radiation and decreased sex ratio of the offspring was not strong. Therefore, while interpreting the results of this study, it is important to note that the results do not support the practice of intentional testicular radiation exposure for gender selection.

## Conclusions

Herein, we reported a possible association between testicular radiation exposure and low male sex ratio of the offspring. Confirmatory evidence is needed from larger studies which measure the pre-conceptional doses accumulated in various temporal periods, separating out spermatogonial and spermatid effects.

## Supporting information

**S1 Dataset. Minimal data set used in the analysis.**
(XLSX)

## Acknowledgments

We would like to thank Editage (https://www.editage.jp/) for English language editing.

## Author Contributions

**Conceptualization:** Yasukazu Hijikata, Masayuki Nakahara, Akira Kusumegi.

**Data curation:** Yasukazu Hijikata, Akira Kusumegi.

**Formal analysis:** Yasukazu Hijikata.

**Investigation:** Yasukazu Hijikata, Akira Kusumegi.

**Methodology:** Yasukazu Hijikata.

**Project administration:** Yuichi Takahashi.

**Software:** Yasukazu Hijikata.

**Supervision:** Nozomi Hatano.

**Writing – original draft:** Yasukazu Hijikata.

**Writing – review & editing:** Masayuki Nakahara, Akira Kusumegi, Junji Morii, Naoki Okubo, Nozomi Hatano, Yuichi Takahashi.

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
