## [Decision Letter · Decision Letter 0]

4 May 2021

PONE-D-21-09546

Association between occupational testicular radiation exposure and lower male sex ratio of offspring among orthopedic surgeons

PLOS ONE

Dear Dr. Hijikata,

Thank you for submitting your manuscript to PLOS ONE. After careful consideration, we feel that it has merit but does not fully meet PLOS ONE’s publication criteria as it currently stands. Therefore, we invite you to submit a revised version of the manuscript that addresses the points raised during the review process.

The manuscript is generally well written, but limitations of the study (e.g., the lack of quantitative radiation dose information for testicular exposure and support of the present findings from other studies) has not explicitly been discussed nor acknowledged in the abstract. The revision should adequately address all the comments raised by the two reviewers.

We look forward to receiving your revised manuscript.

Kind regards,

Nobuyuki Hamada

Academic Editor

PLOS ONE

Journal Requirements:

2. In your ethics statement in the Methods section and in the online submission form, please ensure that you have discussed whether any identifying information was collected in the questionnaire, or whether the researchers had contact with the respondents.

3. Please include your actual numerical p-values in Table 3.

4. In your Methods section, please provide additional information about the participant recruitment method and the demographic details of your participants. Please ensure you have provided sufficient details to replicate the analyses such as:

a) the recruitment date range (month and year),

b) the names of the five institutions that participated,

c) a table of relevant demographic details or doctors if available,

d) a statement as to whether your sample can be considered representative of a larger population, and

e) a description of how participants were recruited.

Reviewers' comments:

Reviewer's Responses to Questions

**Comments to the Author**

1. Is the manuscript technically sound, and do the data support the conclusions?

Reviewer #1: No

Reviewer #2: No

2. Has the statistical analysis been performed appropriately and rigorously? 

Reviewer #1: No

Reviewer #2: Yes

3. Have the authors made all data underlying the findings in their manuscript fully available?

Reviewer #1: No

Reviewer #2: No

4. Is the manuscript presented in an intelligible fashion and written in standard English?

Reviewer #1: Yes

Reviewer #2: No

5. Review Comments to the Author

Reviewer #1: GENERAL COMMENTS TO THE AUTHORS:

This paper describes the results of a survey of full-time male doctors employed in 3 different professions with access to radiation with regards to the sex of their 109 children. The doctors responded to a survey asking questions about work 1 year ahead of child’s birth. The authors speculate that inadequate protection from x-ray radiation from fluoroscopy during orthopedic surgery could have an effect on the sex ratio of children born after radiation exposure. This conclusion is not supported by the data and a giant leap is made from a small survey with absolutely no information on the nature and magnitude of radiation exposures to the conclusion that “results imply an association between testicular radiation exposure and low male sex ratio of offspring. This result might help young orthopedic surgeons to recognize the risks of radiation exposure and to take protective action against it.”

There are multiple factors that could affect the ratio of boys to girls and the biological mechanism for this association has not been proposed in this paper. Before any conclusions are made, it would be necessary to understand the magnitude of radiation exposures in this group of physicians and the amount of scatter radiation to the testes from fluoroscopic procedures during orthopedic surgery.

SPECIFIC COMMENTS TO THE AUTHORS:

INTRODUCTION

Line 60:

“[8-13].”: Ref #12 is not appropriate.

Line 67:

“sex ratio of the offspring”: The authors need to cite literature (if exist) that this concern is justified.

MATERIALS AND METHODS

Line 81:

“a retrospective longitudinal analysis”: Should use standard epidemiological study designs, e.g. cohort study.

Line 87:

“a questionnaire survey”: I am not sure I agree. Even questionnaires require consent. Do you have an IRB approval for this study? After reaching the paper through, I think it is required.

Line 100:

“recall bias”: Recall bias only happens in case-control studies; therefore, this cannot be a longitudinal study.

Line 119:

“lateral and posterior radiation exposure.”: How much of the radiation dose could be received this way? Could you please give us an idea? What is the mean cumulative, badge-based dose for study participants? What percentage of it do you think could be due to scatter radiation.

DISCUSSION

Line 181:

“comprehensive”: Why does this study qualify to be called “comprehensive”?

Line 187:

“who are exposed”: If this is correct, there are bigger concerns about the effects of radiation in those physicians than changing sex ratio.

Line 193:

“89%”: What was the response rate in this study?

Line 246:

“imply a chance effect.”: What were the badge doses?

CONCLUSIONS

Line 252:

This is a case-control study: you started with cases (male offspring) and controls (female offspring) and then interviewed them about place of work 1 year before child’s birth. However, because outcomes are not rare (~50%), odds do not approximate risks. This is most likely due to biased sample selection and recall bias.

REFERENCES

Line 312:

“12.”: It is hard to understand why this reference was chosen. See Ashmore et. al. 2010, which addresses problems with Canadian NDR.

Reviewer #2: General comments

This is an interesting and generally well written examination of sex ratio in offspring of orthopedic surgeons. The findings, although apparently quite strong, are somewhat outweighed by the absence of quantitative dose information. The authors use of the 12 months preceding birth is an odd one. If the sex ratio changes are a spermatid effect then the relevant period is from 12 to 9 months before the birth of a child, while if the changes are a spermatogonial effect than then anything prior to 12 months before birth should be considered. The Discussion is a little skimpy, and the “Comparison with other studies” section could usefully consider the findings in other studies, which are mostly null or in the opposite direction to the present study. I think the authors should tone down the conclusions in the Abstract and at the end of the Discussion in the light of the problems with this study and the lack of support of their findings from other studies. Although the MS is generally clear and well written, the language is occasionally non-idiomatic and would benefit from services of a native English speaker.

Detailed comments (page, line)

p.3 l.48-49 This sentence (“This result …”) should probably be toned down in the light of the problems with this study and the lack of support of these findings from other studies.

p.7 l.130-137 The use of one year before the birth of the child as the relevant period is odd. If the sex ratio changes are a spermatid effect, as the authors seem to hypothesize then the relevant period is from 12 to 9 months before birth (i.e. 3 months before conception), since the process of sperm maturation is about 2-3 months. If spermatogonial effects are more relevant then it would be anything before 12 months before birth (i.e., more than 3 months before conception). The analysis needs to be changed to reflect these biological data.

p.11 l.190-p.12 l.205 The statement “The results of this study are consistent with the available evidence” is not altogether correct. The study of Jablon & Kato (Am J Epidemiol 1971 93 253-8) of the offspring of Japanese atomic bomb survivors found no paternal-radiation-associated change in sex ratio. Dickinson et al (J Epidemiol Commun Health 1996 50 645-52) found some evidence of elevation of male:female sex ratio in offspring of British nuclear workers receiving >10 mSv 90-day preconceptional dose (so in the opposite direction to that implied by this study) but there was no association with total preconceptional dose. Shea et al (Am J Epidemiol 1997 145 546-51) studied a general UK population and found that the sex ratio in exposed fathers was the same as in unexposed fathers; however, there was no dosimetry in this study. Koya et al (Radiat Environ Biophys 2015 54 453-63) observed no change in sex ratio with father’s preconceptional dose in the Kerala high background radiation area of India. Scherb et al (Environ Health 2013 12 63) document a jump in the sex ratio, with increased proportion of male births, post 1986 in Russia and Cuba, which they attribute to radioactive contamination from the Chernobyl nuclear accident; however there was no dosimetry in this study, but nevertheless the results point in the opposite direction to those in the present study. The study of Hama et al that the authors discuss is interesting, but it is based on a self-selected survey sample and the possibility of selection bias cannot be discounted. The very small reduction in sex ratio in the study of Pillarisetti et al is probably not significant. The Discussion section could usefully discuss all these findings.

p.13 l.233-234 This sentence (“First, exposure variables … exposure.”) is somewhat misleading. Quantitative radiation exposure variables are not available in the present study, or at least not given. The only measure of exposure is the type of job performed, which is probably a poor surrogate. It might be best if this is rephrased as “First, the fact of possible radiation exposure may be misclassified, and there is no measurement of actual testicular radiation exposure.”

p.13 l.238-239 The sentence “Therefore it is not possible…” should probably be removed, as it can be inferred from what is said previously, assuming the sentence above is modified in the say suggested.

p.14 l.254-255 This sentence (“This result …”) should probably be toned down in the light of the problems with this study and the lack of support of these findings from other studies.

6. PLOS authors have the option to publish the peer review history of their article (what does this mean?). If published, this will include your full peer review and any attached files.

Reviewer #1: No

Reviewer #2: No

---

## [Author Response · Author response to Decision Letter 0]

18 Jun 2021

Response to Reviewers:

Reviewer #1

Thank you for your important remarks. Our responses to your comments are found below.

GENERAL COMMENTS TO THE AUTHORS:

This paper describes the results of a survey of full-time male doctors employed in 3 different professions with access to radiation with regards to the sex of their 109 children. The doctors responded to a survey asking questions about work 1 year ahead of child’s birth. The authors speculate that inadequate protection from x-ray radiation from fluoroscopy during orthopedic surgery could have an effect on the sex ratio of children born after radiation exposure. This conclusion is not supported by the data and a giant leap is made from a small survey with absolutely no information on the nature and magnitude of radiation exposures to the conclusion that “results imply an association between testicular radiation exposure and low male sex ratio of offspring. This result might help young orthopedic surgeons to recognize the risks of radiation exposure and to take protective action against it.”

There are multiple factors that could affect the ratio of boys to girls and the biological mechanism for this association has not been proposed in this paper. Before any conclusions are made, it would be necessary to understand the magnitude of radiation exposures in this group of physicians and the amount of scatter radiation to the testes from fluoroscopic procedures during orthopedic surgery.

Response

Thank you for your valuable suggestions. As you pointed out, the sample size of this study is small, and the methods are not rigorous; therefore, we understand that we can only report only exploratory results. We have improved the content of the manuscript (particularly the interpretation of the results) and included a review of the existing evidence. We would love to hear from you concerning this.

SPECIFIC COMMENTS TO THE AUTHORS:

INTRODUCTION

Line 60:

“[8-13].”: Ref #12 is not appropriate.

Response

We read the study by Ashore (2010) and realized that there was a problem with the Canadian NDR. Ref #12, Sent (2001), has been deleted as it was cited for background information only and does not relate significantly to the content of the manuscript. Thank you very much for providing the reference. It was very helpful.

Line 67:

“sex ratio of the offspring”: The authors need to cite literature (if exist) that this concern is justified.

Response

Thank you for this plausible suggestion. The paper that prompted us to conduct this study is Zadeh HG (1997), modified Ref #16 (Ref. #17 before revision). This reference has been cited in the manuscript (lines 72-74, page 4 of the 'Manuscript'), and its contents have been presented and compared with the results of the present study in the Discussion (lines 219-221, page 12 of the 'Manuscript').

MATERIALS AND METHODS

Line 81:

“a retrospective longitudinal analysis”: Should use standard epidemiological study designs, e.g. cohort study.

Response

Thank you for your suggestion. Your comment on recall bias below helped us to understand that this was a case-control design. We have changed longitudinal analysis to case-control study (line 83, page 5 of the 'Manuscript').

Line 87:

“a questionnaire survey”: I am not sure I agree. Even questionnaires require consent. Do you have an IRB approval for this study? After reaching the paper through, I think it is required.

Response

Thank you for your question. On June 8, 2021, an additional ethical review was carried out by the certified ethics committee of the main research institution, Shinkomonji hospital, regarding the ethical appropriateness of the analysis and publication of the results using the dataset currently in our possession. The study was approved. We have added this information in the study design and setting (lines 90-91, page 5 of the 'Manuscript').

Line 100:

“recall bias”: Recall bias only happens in case-control studies; therefore, this cannot be a longitudinal study.

Response

Thank you for this remark. We recognized that a case-control design was most appropriate, as the study measured the presence or absence of the outcome (female sex of the child) at the time of the survey and measured the presence or absence of the exposure (opportunity for testicular radiation exposure) retrospectively. As mentioned above, we have revised the study design section (line 83, page 5 of the 'Manuscript').

Line 119:

“lateral and posterior radiation exposure.”: How much of the radiation dose could be received this way? Could you please give us an idea? What is the mean cumulative, badge-based dose for study participants? What percentage of it do you think could be due to scatter radiation.

Response

Thank you for these very important questions. As we mentioned in the Discussion, we are aware that one serious limitation of this study was the inability to measure the actual radiation dose. Since we were unable to provide additional data, we cited the study by Funao et al. (modified ref. 18), wherein the radiation dose was similar to ours (lines 131-136, page 7 of the 'Manuscript').

DISCUSSION

Line 181:

“comprehensive”: Why does this study qualify to be called “comprehensive”?

Response

Thank you for your question. We referred to the study as ‘comprehensive,’ because we had data for the entire cohort, as all subjects in the study settings were included. However, following your remarks, we understood that the study is a case-control study. We have deleted the term ‘comprehensive’ since it is not appropriate in a case-control design i.e., the source cohort is not captured (lines 198, page 11 of the 'Manuscript').

Line 187:

“who are exposed”: If this is correct, there are bigger concerns about the effects of radiation in those physicians than changing sex ratio.

Response

Thank you for pointing this out. The phrase ‘who are exposed’ was clearly an exaggeration. We have replaced it with a more conservative term, ‘who might be exposed.’ There is no doubt that inexperienced doctors (as we were) tend to rely more on radiography in clinical practice, and the results of ref.5 suggest this. To confirm that the younger generation is more exposed to radiation, radiation exposure should be measured. The Society for MIST (https://s-f-mist.com/en/), of which we are members, has taken the lead in conducting a survey to determine the extent to which radiation damage is actually occurring; publication of the results is pending.

Line 193:

“89%”: What was the response rate in this study?

Response

Thank you for your question. Although the sample size was small, the response rate was 100%. It has been stated in the Results that ‘the responses were obtained from all eligible doctors.’ We have also added the percentage of responses (lines 170, page 9 of the 'Manuscript').

Line 246:

“imply a chance effect.”: What were the badge doses?

Response

Thank you for your question. We could not obtain information on the actual radiation dose around the testes for each group, as the rate of badge wearing was very low in the target population, and the badges were never worn around the testes. The term ‘chance effect’ was used because we could not explain the mechanism by which the sex ratio skewed towards males in the RNT and RT groups. Therefore, it was most likely a ‘chance effect.’

CONCLUSIONS

Line 252:

This is a case-control study: you started with cases (male offspring) and controls (female offspring) and then interviewed them about place of work 1 year before child’s birth. However, because outcomes are not rare (~50%), odds do not approximate risks. This is most likely due to biased sample selection and recall bias.

Response

Thank you for these important remarks. As you pointed out, this is a case-control study, and the outcomes are not rare. Therefore, the odds ratio cannot be interpreted as a risk ratio. This means that therefore, we cannot conclude from the results that ‘the RT group was four times more likely to have daughters.’ However, we believe that the odds ratios were useful in determining whether there was an association. We also believe that there was no significant selection bias, at least in the target population, as all respondents were available at the time of the survey. Furthermore, although participants had to recall information from up to 11 years prior to the study, it was easy for them to recall their department at that time; therefore, the recall bias was small.

 The manuscript has been revised substantially to bring it closer to a reasonable interpretation. First, the Key results and clinical implications were revised for a better interpretation of the results (lines 198-208, page 11-12 of the 'Manuscript'). Second, to avoid misunderstanding about the odds ratios, we have added a limitation to the Strengths and limitations section (lines 289-293, page 15 of the 'Manuscript'). Finally, the claims made in the Conclusions have been modified (lines 51, page 3 and, lines 303-304, page 16 of the 'Manuscript').

REFERENCES

Line 312:

“12.”: It is hard to understand why this reference was chosen. See Ashmore et. al. 2010, which addresses problems with Canadian NDR.

Response

Ref #12 has been deleted. Thank you for pointing this out.

 

Reviewer #2

Thank you for your valuable suggestions that have helped us improve our manuscript. Our responses to each of your comments are presented below.

General comments

This is an interesting and generally well written examination of sex ratio in offspring of orthopedic surgeons. The findings, although apparently quite strong, are somewhat outweighed by the absence of quantitative dose information. The authors use of the 12 months preceding birth is an odd one. If the sex ratio changes are a spermatid effect then the relevant period is from 12 to 9 months before the birth of a child, while if the changes are a spermatogonial effect than then anything prior to 12 months before birth should be considered. The Discussion is a little skimpy, and the “Comparison with other studies” section could usefully consider the findings in other studies, which are mostly null or in the opposite direction to the present study. I think the authors should tone down the conclusions in the Abstract and at the end of the Discussion in the light of the problems with this study and the lack of support of their findings from other studies. Although the MS is generally clear and well written, the language is occasionally non-idiomatic and would benefit from services of a native English speaker.

Response

Thank you very much for summarizing the results and highlighting the issues. We will respond to your comments in more detail below. This first submitted version of the manuscript was edited by Editage (https://www.editage.jp/). We sent it to Editage for proofreading again.

Detailed comments (page, line)

p.3 l.48-49 This sentence (“This result …”) should probably be toned down in the light of the problems with this study and the lack of support of these findings from other studies.

Response

Thank you for pointing this out. The interpretation of our results was exaggerated. We have substantially revised the Key results and clinical implications (lines 198-208, page 11-12 of the 'Manuscript') and Conclusions (lines 51, page 3 and lines 303-304, page 16 of the 'Manuscript') to understate the interpretation of the results.

p.7 l.130-137 The use of one year before the birth of the child as the relevant period is odd. If the sex ratio changes are a spermatid effect, as the authors seem to hypothesize then the relevant period is from 12 to 9 months before birth (i.e. 3 months before conception), since the process of sperm maturation is about 2-3 months. If spermatogonial effects are more relevant then it would be anything before 12 months before birth (i.e., more than 3 months before conception). The analysis needs to be changed to reflect these biological data.

Response

Thank you very much for your valuable comments. As you pointed out, we stated the rationale for using the one year cut-off poorly, and the inaccuracy of the rationale was not fully explained. We hypothesized that the period at risk was the period after the secondary spermatocyte formation, when the sperm carries either the X or Y chromosome. AS you pointed out, the period at risk is not a time point, but a period with a range of 2–3 months (the period during which the secondary spermatocyte becomes sperm and is stored in the epididymis). Therefore, limiting the measurement of an exposure to a single time point may lead to misclassification. However, we believe that the magnitude of misclassification was small because it is rare for a personnel’s department affiliation to be changed within 2–3 months. Furthermore, limiting the exposure measurement to ‘one year ago’ makes it easier to be explained. Most parent do not forgets their child's birthday; therefore, we thought that it would be very easy to answer the question: ‘What department were you in exactly one year before your child's birthday?.’ We are unsure whether this was responsible for the 100% response rate.

We have added more accurate information about spermatogenesis in the Outcome of interest and exposure section (lines 143-153,　page 7-8 of the 'Manuscript'). In addition, the possibility of a misclassification of the exposure has been added to the limitations (lines 280-284, page 15 of the 'Manuscript').

p.11 l.190-p.12 l.205 The statement “The results of this study are consistent with the available evidence” is not altogether correct. The study of Jablon & Kato (Am J Epidemiol 1971 93 253-8) of the offspring of Japanese atomic bomb survivors found no paternal-radiation-associated change in sex ratio. Dickinson et al (J Epidemiol Commun Health 1996 50 645-52) found some evidence of elevation of male:female sex ratio in offspring of British nuclear workers receiving >10 mSv 90-day preconceptional dose (so in the opposite direction to that implied by this study) but there was no association with total preconceptional dose. Shea et al (Am J Epidemiol 1997 145 546-51) studied a general UK population and found that the sex ratio in exposed fathers was the same as in unexposed fathers; however, there was no dosimetry in this study. Koya et al (Radiat Environ Biophys 2015 54 453-63) observed no change in sex ratio with father’s preconceptional dose in the Kerala high background radiation area of India. Scherb et al (Environ Health 2013 12 63) document a jump in the sex ratio, with increased proportion of male births, post 1986 in Russia and Cuba, which they attribute to radioactive contamination from the Chernobyl nuclear accident; however there was no dosimetry in this study, but nevertheless the results point in the opposite direction to those in the present study. The study of Hama et al that the authors discuss is interesting, but it is based on a self-selected survey sample and the possibility of selection bias cannot be discounted. The very small reduction in sex ratio in the study of Pillarisetti et al is probably not significant. The Discussion section could usefully discuss all these findings.

Response

Thank you very much for all the reference-by-reference information you have provided. In the manuscript, we only cited previous studies on occupational exposure to medical X-rays among healthcare workers. We have reviewed the papers you suggested and have made significant additions and revisions to the Comparison with other studies section for a more in-depth discussion (lines 228-247, page 13 of the 'Manuscript').

p.13 l.233-234 This sentence (“First, exposure variables … exposure.”) is somewhat misleading. Quantitative radiation exposure variables are not available in the present study, or at least not given. The only measure of exposure is the type of job performed, which is probably a poor surrogate. It might be best if this is rephrased as “First, the fact of possible radiation exposure may be misclassified, and there is no measurement of actual testicular radiation exposure.”

Response

Thank you for your detailed remarks. We have revised the sentence (lines 275-277, page 15 of the 'Manuscript').

p.13 l.238-239 The sentence “Therefore it is not possible…” should probably be removed, as it can be inferred from what is said previously, assuming the sentence above is modified in the say suggested.

Response

We have deleted the sentence as suggested.

p.14 l.254-255 This sentence (“This result …”) should probably be toned down in the light of the problems with this study and the lack of support of these findings from other studies.

Response

Following your remarks, we have toned down our conclusions. Thank you very much for reading the manuscript in detail and for your constructive comments. We have revised in the Conclusions (lines 51, page 3 and, lines 303-304, page 16 of the 'Manuscript').

---

## [Decision Letter · Decision Letter 1]

9 Aug 2021

PONE-D-21-09546R1

Association between occupational testicular radiation exposure and lower male sex ratio of offspring among orthopedic surgeons

PLOS ONE

Dear Dr. Hijikata,

Thank you for submitting your manuscript to PLOS ONE. After careful consideration, we feel that it has merit but does not fully meet PLOS ONE’s publication criteria as it currently stands. Therefore, we invite you to submit a revised version of the manuscript that addresses the points raised during the review process.

Please adequately address all the comments raised by the reviewer. Your second revision will be sent out to another round of review.

We look forward to receiving your revised manuscript.

Kind regards,

Nobuyuki Hamada

Academic Editor

PLOS ONE

Reviewers' comments:

Reviewer's Responses to Questions

**Comments to the Author**

1. If the authors have adequately addressed your comments raised in a previous round of review and you feel that this manuscript is now acceptable for publication, you may indicate that here to bypass the “Comments to the Author” section, enter your conflict of interest statement in the “Confidential to Editor” section, and submit your "Accept" recommendation.

Reviewer #2: (No Response)

2. Is the manuscript technically sound, and do the data support the conclusions?

Reviewer #2: Partly

3. Has the statistical analysis been performed appropriately and rigorously? 

Reviewer #2: Yes

4. Have the authors made all data underlying the findings in their manuscript fully available?

Reviewer #2: Yes

5. Is the manuscript presented in an intelligible fashion and written in standard English?

Reviewer #2: Yes

6. Review Comments to the Author

Reviewer #2: General comments

This is an interesting and generally well written examination of sex ratio in offspring of orthopedic surgeons. The paper is much improved on the first version. However, there are still quite a few weaknesses, both in the presentation of the Tables and particularly in the Discussion.

Detailed comments (page, line)

p.3 l.51 I would replace the final sentence by “Confirmatory evidence is needed from larger studies which measure the pre-conceptional doses accumulated in various temporal periods, separating out spermatogonial and spermatid effects.”.

p.8 l.146-153 What evidence is there in support of this hypothesis, that sex ratio results entirely from spermatid effects? If this is (as I suspect) a largely post hoc hypothesis then this should be made clear.

p.8 l.148 I assume that by “2-3 months” is meant “2-3 months before conception”.

p.8 l.151-153 Is this sentence (“This means that…”) correct? Surely it is the period 1-3 months before conception rather than the date of birth that is meant.

Tables 1, 2, 3 It is needlessly confusing that in Table 1 the proportion of female children is given, whereas in Table 2 it is effectively the proportion of male children that is given. Although the labelling of Table 3 does not make this clear, it must be the case that it is the odds ratio of female:male births that is given. The labelling of Table 3 must be improved, and all three Tables use more or less the same measure (males/total or females/total, it does not matter which).

p.12 l.211 – p.13 l.227 The sentence on p.12 l.211-212 (“The results of this study are consistent …”) should be removed or at least toned down. The evidence cited in support of this statement in the rest of the para is really rather weak. As previously noted by the referee the study of Hama et al that the authors discuss is interesting, but it is based on a self-selected survey sample and the possibility of selection bias cannot be discounted. The very small reduction in sex ratio in the study of Pillarisetti et al is probably not significant. These weaknesses and limitations of this evidence must be discussed here.

p.13 l.224-227 This sentence (“The results of this study, obtained …”) repeats what is said at the beginning of the Discussion, and should be removed.

p.13 l.246-247 This sentence (“Therefore, it should be noted …”) is rather odd. Are the authors really implying that the effects observed are not due to the radiation alone, but to some other occupational factor specific to healthcare workers? Obviously if the effects are related to radiation exposure (rather than some unknown factor specific to this occupational group) there is no reason why they should not apply to all radiation exposed groups. If the findings relate to some factor specific to this occupational group then this considerably lessens their scientific interest, and begs the question as to whether radiation exposure is really the cause of what has been observed in this study.

p.15 l.276-277 There should be discussion at this point of the limitations of the data in relation to timing of exposure. In particular the data make it quite difficult to differentiate between spermatid effects (which the authors assume, as above, but possibly based on no strong prior body of data) and spermatogonial effects. This hypothesis could also be discussed.

p.15 l.293 – p.16 l.294 The sentence is not terribly clear, and the authors should also highlight the role that confounding by some unknown factor may play. So perhaps rephrase as: “Finally, although the results were highly statistically significant, they are based on a quite small sample, and the play of chance, or some unidentified confounding factor, cannot be ruled out.”.

p.16 l.303-304 I would replace the final sentence by “Confirmatory evidence is needed from larger studies which measure the pre-conceptional doses accumulated in various temporal periods, separating out spermatogonial and spermatid effects.”.

7. PLOS authors have the option to publish the peer review history of their article (what does this mean?). If published, this will include your full peer review and any attached files.

Reviewer #2: No

---

## [Author Response · Author response to Decision Letter 1]

21 Aug 2021

Response to Reviewers:

Reviewer #2

General comments

This is an interesting and generally well written examination of sex ratio in offspring of orthopedic surgeons. The paper is much improved on the first version. However, there are still quite a few weaknesses, both in the presentation of the Tables and particularly in the Discussion. 

Response: Thank you for your insightful comments. We have revised the manuscript in line with your suggestions. We hope that our edits and the responses we have provided below satisfactorily address all the issues and concerns you have noted and that the revised manuscript is now suitable for publication.

Detailed comments (page, line)

p.3 l.51 

I would replace the final sentence by “Confirmatory evidence is needed from larger studies which measure the pre-conceptional doses accumulated in various temporal periods, separating out spermatogonial and spermatid effects.”. 

Response: Thank you for this comment. We have accordingly replaced the final sentence (p.3 l.47-49).

Revise: Confirmatory evidence is needed from larger studies which measure the pre-conceptional doses accumulated in various temporal periods, separating out spermatogonial and spermatid effects.

p.8 l.146-153

What evidence is there in support of this hypothesis, that sex ratio results entirely from spermatid effects? If this is (as I suspect) a largely post hoc hypothesis then this should be made clear. 

Response: Thank you for your comment. To clarify, this is not a post-hoc hypothesis. There is a widespread rumor among orthopeadic surgeons in Japan that the children of orthopedic surgeons tend to be female. If this is indeed true, we would like to understand why this is so. Thus, we formulated this hypothesis and designed this study to test it. However, as you have pointed out, this hypothesis has no supportive evidence, and we have accordingly made this clear (p.7 l.141).

Revise: We hypothesized, albeit without supporting evidence, that…

p.8 l.148

I assume that by “2-3 months” is meant “2-3 months before conception”. 

Response: Yes, and per your comment, we have revised this in the manuscript for clarity (p.8 l.143-144).

Revise: 2–3 months before conception.

p.8 l.151-153

Is this sentence (“This means that…”) correct? Surely it is the period 1-3 months before conception rather than the date of birth that is meant.

Response: We apologize for the unclear text. We have now corrected this to "The risk period began one year before the child’s birthday” (p.8 l.146-149).

Revise: In other words, considering that the gestation period is approximately 280 days, the period at risk of a decrease in the male sex ratio of the child due to testicular radiation exposure began approximately one year before the child's birthday.

Tables 1, 2, 3

It is needlessly confusing that in Table 1 the proportion of female children is given, whereas in Table 2 it is effectively the proportion of male children that is given. Although the labelling of Table 3 does not make this clear, it must be the case that it is the odds ratio of female:male births that is given. The labelling of Table 3 must be improved, and all three Tables use more or less the same measure (males/total or females/total, it does not matter which). 

Response: Thank you very much for your meaningful comments. We apologize for the confusing presentation of our tables. Per your comment, we have presented all calculated sex ratios in this study as female sex ratio and have accordingly revised the relevant sections (i.e., Abstract [p.2 l.33-34 and l.41], Mateials and Methods [p.8 l.155-156], Results [p.10 l.183, l.185], and Tables 1 and 2). Moreover, in other instances, we have clearly indicated whether we are referring to female or male sex ratio (p.4 l.70, p.5 l.75, p.7 l.139, l.142, p.8 l.148 and l.155). We have also revised the title of Table 3 to clearly indicate that the odds ratios of having female children are presented (p.11 l.207).

p.12 l.211 – p.13 l.227

The sentence on p.12 l.211-212 (“The results of this study are consistent …”) should be removed or at least toned down. The evidence cited in support of this statement in the rest of the para is really rather weak. As previously noted by the referee the study of Hama et al that the authors discuss is interesting, but it is based on a self-selected survey sample and the possibility of selection bias cannot be discounted. The very small reduction in sex ratio in the study of Pillarisetti et al is probably not significant. These weaknesses and limitations of this evidence must be discussed here.

Response: Thank you for your significant remarks. We have revised the tone of this particular sentence (p.12 l.234) and corrected the last sentence of the paragraph (p.13 l.245-246).

Revise: Our results are consistent with the inconclusive evidence…

However, these results are limited by selection biases because they used questionnaires with voluntary responses.

p.13 l.224-227 

This sentence (“The results of this study, obtained …”) repeats what is said at the beginning of the Discussion, and should be removed.

Response: We have deleted this sentence per your comment.

p.13 l.246-247

This sentence (“Therefore, it should be noted …”) is rather odd. Are the authors really implying that the effects observed are not due to the radiation alone, but to some other occupational factor specific to healthcare workers? Obviously if the effects are related to radiation exposure (rather than some unknown factor specific to this occupational group) there is no reason why they should not apply to all radiation exposed groups. If the findings relate to some factor specific to this occupational group then this considerably lessens their scientific interest, and begs the question as to whether radiation exposure is really the cause of what has been observed in this study.

Response: Thank you for pointing this out. We believe that radiation affects not only medical professionals but also the general population. We understand that our phrasing was misleading and have deleted the sentence accordingly.

p.15 l.276-277

There should be discussion at this point of the limitations of the data in relation to timing of exposure. In particular the data make it quite difficult to differentiate between spermatid effects (which the authors assume, as above, but possibly based on no strong prior body of data) and spermatogonial effects. This hypothesis could also be discussed.

Response: We appreciate your comments. As you have pointed out, we have discussed the limitations of our data in relation to timing exposure (p.15 l.301-303).

Revise: In addition, because we could not determine the timing of exposure, we cannot differentiate between spermatid effects, which we hypothesized, and spermatogonial effects.

p.15 l.293 – p.16 l.294

The sentence is not terribly clear, and the authors should also highlight the role that confounding by some unknown factor may play. So perhaps rephrase as: “Finally, although the results were highly statistically significant, they are based on a quite small sample, and the play of chance, or some unidentified confounding factor, cannot be ruled out.”. 

Response: Per your comment, we have rephrased this for clarity (p.16 l.312-314).

Revise: Finally, although the results were highly statistically significant, they are based on a quite small sample, and the play of chance, or some unidentified confounding factor, cannot be ruled out.

p.16 l.303-304

I would replace the final sentence by “Confirmatory evidence is needed from larger studies which measure the pre-conceptional doses accumulated in various temporal periods, separating out spermatogonial and spermatid effects.”.

Response: Thank you for this suggestion. We have accordingly revised this (p.16 l.322-324).

Revise: Confirmatory evidence is needed from larger studies which measure the pre-conceptional doses accumulated in various temporal periods, separating out spermatogonial and spermatid effects.

---

## [Decision Letter · Decision Letter 2]

19 Dec 2021

Association between occupational testicular radiation exposure and lower male sex ratio of offspring among orthopedic surgeons

PONE-D-21-09546R2

Dear Dr. Hijikata,

We’re pleased to inform you that your manuscript has been judged scientifically suitable for publication and will be formally accepted for publication once it meets all outstanding technical requirements.

Kind regards,

Nobuyuki Hamada

Academic Editor

PLOS ONE

Additional Editor Comments (optional):

Reviewers' comments:

Reviewer's Responses to Questions

**Comments to the Author**

1. If the authors have adequately addressed your comments raised in a previous round of review and you feel that this manuscript is now acceptable for publication, you may indicate that here to bypass the “Comments to the Author” section, enter your conflict of interest statement in the “Confidential to Editor” section, and submit your "Accept" recommendation.

Reviewer #2: All comments have been addressed

2. Is the manuscript technically sound, and do the data support the conclusions?

Reviewer #2: Yes

3. Has the statistical analysis been performed appropriately and rigorously? 

Reviewer #2: Yes

4. Have the authors made all data underlying the findings in their manuscript fully available?

Reviewer #2: No

5. Is the manuscript presented in an intelligible fashion and written in standard English?

Reviewer #2: Yes

6. Review Comments to the Author

Reviewer #2: General comments

This is an interesting and generally well written examination of sex ratio in offspring of orthopedic surgeons. The paper is much improved on the second version, and has met all my concerns.

7. PLOS authors have the option to publish the peer review history of their article (what does this mean?). If published, this will include your full peer review and any attached files.

Reviewer #2: No

---

## [Editor Report · Acceptance letter]

21 Dec 2021

PONE-D-21-09546R2 

Association between occupational testicular radiation exposure and lower male sex ratio of offspring among orthopedic surgeons 

Dear Dr. Hijikata:

I'm pleased to inform you that your manuscript has been deemed suitable for publication in PLOS ONE. Congratulations! Your manuscript is now with our production department. 

Kind regards, 

on behalf of

Dr. Nobuyuki Hamada 

Academic Editor

PLOS ONE